# Contrasting Asymptomatic and Drug Resistance Gene Prevalence of *Plasmodium falciparum* in Ghana: Implications on Seasonal Malaria Chemoprevention

**DOI:** 10.3390/genes10070538

**Published:** 2019-07-16

**Authors:** Cheikh Cambel Dieng, Lauren Gonzalez, Kareen Pestana, Shittu B. Dhikrullahi, Linda E. Amoah, Yaw A. Afrane, Eugenia Lo

**Affiliations:** 1Department of Biological Sciences, University of North Carolina at Charlotte, Charlotte, NC 28223, USA; 2Department of Medical Microbiology, College of Health Sciences, University of Ghana, P.O. Box LG 25, Accra, Ghana; 3Noguchi Memorial Institute for Medical Research, University of Ghana, P.O. Box LG 25, Accra, Ghana

**Keywords:** *Plasmodium falciparum*, *pfdhps*, *phdhfr*, asymptomatic infections, antimalarial drug resistance, Sulfadoxine-Pyrimethamine, quantitative real-time PCR, TARE-2

## Abstract

Malaria is a significant public health problem in Ghana. Seasonal Malaria Chemoprevention (SMC) using a combination of sulfadoxine-pyrimethamine and amodiaquine has been implemented since 2015 in northern Ghana where malaria transmission is intense and seasonal. In this study, we estimated the prevalence of asymptomatic *P. falciparum* carriers in three ecological zones of Ghana, and compared the sensitivity and specificity of different molecular methods in identifying asymptomatic infections. Moreover, we examined the frequency of mutations in *pfcrt*, *pfmdr1*, *pfdhfr*, and *pfdhps* that relate to the ongoing SMC. A total of 535 asymptomatic schoolchildren were screened by microscopy and PCR (18s rRNA and TARE-2) methods. Among all samples, 28.6% were detected as positive by 18S nested PCR, whereas 19.6% were detected by microscopy. A high PCR-based asymptomatic prevalence was observed in the north (51%) compared to in the central (27.8%) and south (16.9%). The prevalence of *pfdhfr*-N51I/C59R/S108N/*pfdhps*-A437G quadruple mutant associated with sulfadoxine-pyrimethamine resistance was significantly higher in the north where SMC was implemented. Compared to 18S rRNA, TARE-2 serves as a more sensitive molecular marker for detecting submicroscopic asymptomatic infections in high and low transmission settings. These findings establish a baseline for monitoring *P. falciparum* prevalence and resistance in response to SMC over time.

## 1. Introduction

In 2018, an estimated 219 million cases of malaria occurred worldwide. Among them, 435,000 malaria-related deaths and 92% of the global malaria burden occurred in sub-Saharan Africa [1,2]. *Plasmodium falciparum* is responsible for the majority of malaria morbidity and mortality. In the last decades, a number of chemotherapeutic agents and insecticides have been used in the control of transmission. However, malaria still remains endemic in many parts of Africa. The rapidly emerging resistance of both the vector mosquitoes and *P. falciparum* has severely impacted the effectiveness of control measures. In addition, socio-economic factors such as poverty and poor healthcare infrastructure, as well as insufficient educational and monitoring systems, make malaria elimination challenging especially in remote areas of many African countries. 

In Ghana, malaria is a significant public health problem, especially in rural areas. The President’s Malaria Initiative, Ghana has employed various control strategies to reduce malaria infection. These include free malaria diagnosis and treatment across the country, as well as routine indoor sprays of insecticides against mosquito vectors and distribution of free long-lasting insecticidal bednets in the north [3]. These strategies have resulted in striking reductions in a number of clinical cases. The percentage of outpatient attendance in public health facilities decreased sharply from 48% in 2008 to 28% in 2016 [4]. However, malaria prevalence still remains high in some parts of the country. A multiple cluster survey conducted in 2011 has shown a wide variation in parasite prevalence between the north (51%) and south (4%) of Ghana [5]. Such a variation could be explained by a combination of factors such as vector abundance and distribution, climate, environment, land use change, socioeconomic difference, and human movement. For instance, in the coastal (south) and forest (central) areas, malaria transmission is perennial, whereas in the savannah zone (north), malaria transmission is highly seasonal and intense during/after the rainy season (June–October) [6]. The contrasting pattern of malaria transmission plays a key role in determining and implementing malaria control measures in Ghana. 

In highly malaria-endemic areas of Ghana, individuals are frequently exposed to malaria parasites and acquire protective immunity. These individuals provide a parasite reservoir and can initiate transmission [7]. Due to low parasite density, asymptomatic infections are usually less detectable by microscopy or rapid diagnostic tests. Though less sensitive, microscopy is the first-line diagnostic tool for malaria in many developing countries because of its convenience and affordability. Previous studies have shown that over 20% of microscopic-negative samples were positive when diagnosed by PCR [8]. In most cases, asymptomatic individuals are left undiagnosed and untreated, which severely impedes malaria control and elimination. A conventional molecular method such as 18S rRNA nested PCR has been shown to increase sensitivity of detecting submicroscopic infections [9]. Recently, an ultra-sensitive marker Telomere Associated Repetitive Element 2 (TARE-2) has demonstrated better performance than the conventional 18S rRNA marker in detecting low density parasite infections, especially in active case surveillance [10]. TARE-2 is a high-copy telomere-associated repetitive element that has ~250 copies per genome [11], considerably more abundant than the 18S rRNA with only 5–8 copies per genome in *P. falciparum* [12]. This marker has been previously shown to be useful for detecting ultra-low-density *P. falciparum* infections in Papua New Guinea and Tanzania [10]. It was our goal to determine the most sensitive method for detecting asymptomatic infections in high transmission areas of Ghana in this study.

For decades, chloroquine (CQ) had been used as the first-line treatment for malaria in Ghana. However, the emergence and spread of CQ resistance led to the introduction of artemisinin-based combination therapy (ACT) [13]. Currently, the first line of drug for the treatment of non-severe *P. falciparum* malaria in Ghana is the combination of artesunate and amodiaquine [14]. In March 2012, the World Health Organization (WHO) recommended a new intervention against *P. falciparum* malaria in children during the peak transmission season known as Seasonal Malaria Chemoprevention (SMC), where children under five-years-old are given a single dose of sulfadoxine-pyrimethamine (SP) combined with a 3-day course of amodiaquine (AQ) once a month for up to 4 months [15]. SMC has been introduced in 12 African countries since its first announcement. In Burkina Faso, SMC have been shown to be effective in reducing the prevalence of malaria and anemia among children, as well as the occurrence of fever episodes [16]. In Ghana, considering the variation in transmission pattern between the northern savannah and the southern coastal region, the authorities implemented SMC in the north in July 2015 [5,17]. A previous report on sulfadoxine-pyrimethamine resistance in Ghana was documented in 1988, long before the introduction of SMC [18]. It is yet unclear as to whether the SMC has imposed any impact or selective pressure on the parasite genomes. 

It has been well documented that antimalarial resistance is tightly associated with specific mutations in the *P. falciparum* genome. For instance, mutation at *P. falciparum* chloroquine resistance transporter (*Pfcrt*) codon 76T is associated with CQ resistance [19]. Mutation at multidrug resistance transporter (*Pfmdr1*) codons 86Y and 184F is associated with AQ resistance [13]. Mutations in *P. falciparum* dihydrofolate reductase (*Pfdhfr*) have been known for several years to decrease in vitro and clinical *P. falciparum* susceptibility to pyrimethamine [20]. The three mutations N51I, C59R, and S108N in combination are referred to as *pfdhfr* triplet. Sulfadoxine is associated with mutations at codons N51I, C59R, S108N in *pfdhfr*, and pyrimethamine is associated with mutations at A437G and K540E mutations in the dihydropteroate synthase (*pfdhps*) gene. The latter two mutations together are referred to as *pfdhps* double [21]. The combination of *pfdhfr* triple and *pfdhps* double are known as the quintuple mutation. In vivo and in vitro experiments have shown that SP resistance is tightly correlated to *pfdhfr* triple and *pfdhps* double mutations [22].

In this study, we (1) determined the asymptomatic prevalence of *P. falciparum* across the three ecological zones, i.e., the northern savannah, central forest, and southern coast of Ghana; (2) compared the sensitivity and specificity of different molecular methods in order to provide an accurate estimate of asymptomatic prevalence especially in low transmission settings and; (3) examined the frequency of *Pfcrt*, *Pfmdr1*, *Pfdhfr*, and *Pfdhps* mutations that relates to the antimalarial drugs used in the ongoing SMC. These findings will establish a baseline for monitoring *P. falciparum* prevalence and resistance in response to SMC over time.

## 2. Materials and Methods

### 2.1. Ethics Statement

Scientific and ethical approval was given by the Institutional Scientific and Ethical Review boards of the Noguchi Memorial Institute of Medical Research, University of Ghana, Ghana and the University of North Carolina at Charlotte, USA (IRB00001276). Written informed consent/assent for study participation was obtained from all consenting parents/guardians (for minors under the age of 18), and each individual who was willing to participate in the study. All methods were reviewed and approved by the institutional review board (IRB) and performed in accordance with the relevant guidelines and regulations stated in the IRB protocols.

### 2.2. Sampling and Study Sites

Five sites were selected from three ecological zones of Ghana. These included Pagaza (PZ) in Tamale Municipality and Kpalsogou (KG) in Kumbungu district in the northern savannah region; Duase (KD) in Konongo district in the central forest region, and Ada (AD) and Dodowa (DO) in the southern coastal region. Sample collection was conducted during June-September of 2017. Finger-prick blood samples were collected from 535 schoolchildren aged 5 to 14 years old who present no fever or malaria-related symptoms at the time of collection. Thick and thin blood smears were prepared for microscopic examination and 30–50 μL of blood was blotted on Whatman 3MM filter papers. Filter papers were air-dried and stored in zip-sealed plastic bags with silica gel absorbent at room temperature until DNA extraction.

### 2.3. Microscopic and Molecular Screening

Blood smears were examined at a magnification of 100x under microscopes. Parasites were counted against 200 leukocytes and a smear was considered negative when no parasites were observed after counting at least 100 microscopic fields. The density of parasites (parasitaemia) was expressed as the number of asexual *P. falciparum* per microliter of blood, assuming a leukocyte count of 8000/μL. 

Parasite DNA was extracted from dried blood spots by the Saponin/Chelex method [23]. The final extracted volume was 200 μL. Molecular screening of *P. falciparum* was diagnosed by three PCR assays. A nested PCR approach of the *P. falciparum* 18S rRNA gene using previously published primers was used (Appendix A). In addition, the SYBR Green quantitative real-time PCR (qPCR) assays of the 18S rRNA (5–8 copies per genome [12] and the high-copy telomere-associated repetitive element 2 TARE-2 (~250 copies per genome) [11] were used to screen all the samples following the published protocols. DNA from *P. falciparum* isolates 7G8 (MRA-926) and HB3 (MRA-155) were used as positive controls in all amplifications, and water and uninfected samples were used as a negative control to ensure a lack of contamination. Samples yielding a threshold cycle (Ct) higher or equal to 40 were considered negative for *P. falciparum*. The parasite density in a sample was quantified based on the threshold cycle using the following equation: 2 ^E×(40-Ct sample)^, where Ct represents the threshold cycle of the sample and E, the amplification efficiency. The parasite density of samples in each site was reported as geometric mean and range values.

### 2.4. Assessing Sensitivity and Specificity

To evaluate the sensitivity and specificity of the different diagnostic methods in detecting positive asymptomatic samples, a set of 135 samples from the northern savannah sites (KG and PZ) were used. These samples were selected for marker sensitivity comparisons because they were collected from a highly endemic region where asymptomatic infections were expected to be high. The 135 samples were first screened by microscopy, and then 18S rRNA nested PCR followed by 18S rRNA qPCR and TARE-2 qPCR methods. The percentages of positive and negative infections were recorded and compared among these methods. Sensitivity and specificity tests were performed with MedCalc software (version 12; Mariakerke, Belgium). TARE-2 was used as the gold standard given that this gene contains ~250 copies in the *P. falciparum* genome and has been previously shown to be an ultra-sensitive marker in detecting low density *P. falciparum* infections [11]. The sensitivity values were calculated as true positives/(true positive + false negatives), and specificity values were calculated as true negatives/(true negatives + false positives).

### 2.5. Resistance gene Mutation Frequency

Polymorphisms in the *Pfcrt* and *Pfmdr1* were assessed by nested PCR using previously published primers and protocols [24]. A PCR-restriction fragment length polymorphism assay was then used to assess the mutations at codon 76 of *Pfcrt*. Digestion fragments were visualized on a 2% agarose gel. For codons 86 and 184 of *Pfmdr1*, PCR amplicons (603 bp) were purified and sequenced in an ABI 3730xl DNA analyzer. Likewise, the *Pfdhfr* and *Pfdhps* genes were amplified and sequenced using previously published protocols [25]. Mutations at *Pfdhfr* codons 57, 59, and 108, as well as *Pfdhps* codons 437 and 540 were assessed. All sequences were aligned against the *P. falciparum* reference 3D7 strain using Bioedit Sequence Alignment Editor [26]. Chi-square test was used to test for significant differences in mutation prevalence for each of the gene codons among study sites. All statistical analyses were performed in GraphPad Prism 5.0 (GraphPad Software, La Jolla, CA, USA).

## 3. Results

### 3.1. Asymptomic Prevalence across Ecological Zones

Of the 535 total samples, 105 (19.6%) were found positive by microscopy, 153 (28.6%) by 18S nested PCR, and 169 (31.6%) by 18S qPCR methods. In the north (sites KG and PZ), 24 out of 112 (21.4%) samples were microscopic-positive. The number of positive infections increased to 58 (51.8%) when screened by 18S nested PCR method (Figure 1; Appendix A). In the central (site KD), 52 of 216 (24%) samples were microscopic-positive and the number of positive infections increased to 60 (27.8%) by nested PCR. In the south (sites (DO and ADA), 29 of 207 (14%) samples were microscopic-positive. The number of positive infections increased to 35 (16.9%) when screened by nested PCR (Figure 1). The asymptomatic prevalence of infections was markedly different among the three zones, with the north being the highest and the south being the lowest. However, there was no significant difference in the parasite density measured by 18S qPCR method among the positive infections from all sites (*p* = 0.43; Figure 2).

#### 3.1.1. Sensitivity and Specificity of Diagnostic Methods

Among the 135 samples collected from the north (sites KG and PZ), 35 of the samples were detected as positive by microscopy, 62 by 18s nested PCR, 76 by 18s qPCR, and 89 by TARE-2 qPCR (Table 1). Samples screened with 18s nested PCR performed with a sensitivity of 63.8% (95% confidence interval [CI]; range, 53.3 to 73.5%) and a specificity of 91.30% (95%CI; range 71.9 to 98.9%) compared to microscopy. The 18s qPCR and TARE-2 qPCR indicated a considerably higher level of sensitivity (80.9 and 81.2%, respectively), though both methods also yielded a slightly lower specificity than the 18s nested PCR (Table 1). 

#### 3.1.2. Frequency of Resistance Gene Mutations

Out of all the 165 samples that were successfully genotyped for *pfcrt* codon K76T, four (1%) were found to contain the mutant allele 76T. The remaining samples had the wild type (Table 2, Appendix A). No significant difference was detected in the frequency of the *Pfcrt* K76T mutation among the three ecological zones. For *pfmdr1*, a total of 399 samples were successfully genotyped. Of the 120 samples in the northern savannah (sites PZ, KG), 86Y and 184F mutations were found in four (3.3%) and 32 samples (26.7%), respectively. In the central (KD), there were no 86Y mutations and 34 out of 47 samples (72%) harbored the 184F mutation. In the south (DO and AD), five of the 51 samples (10%) had the 86Y mutation and 16 (50%) had the 184F mutation. There was no significant difference in the 86Y and 184F mutations across the three ecological zones. For *pfdhps* the 437G mutant allele was highly dominant in all regions (100% in the south and central; 91–100% in the north; Table 2), with an overall prevalence of 97% across the country. By contrast, the 540E mutation was only found in a single sample in the north (PZ). For *pfdhfr*, the 51I mutant was found in 89% of the samples from the south, 88% in the central region and 90% of the samples in the north. The 59R mutation was found in all the samples from the south and 92% of the samples from the central region. By contrast, only 75% of the samples from the north harbored this mutation. The 108N mutation was found in all the samples from the south, 94% of the samples in the central region, and 98% of samples in the north, making it the most prevalent (98%) mutation across the country (Table 2). 

Across Africa, the prevalence of mutations related to anti-malarial drug resistance also indicated a wide range of variation (Table 3). For *Pfcrt*, the prevalence of the 76T mutation was apparently higher in other parts of Africa (ranged from 43 to 80%; Table 3) compared to our findings in Ghana (Table 2; 1%). For *Pfmdr1*, the prevalence of 86Y mutation was relatively low in Ghana (4%), Gambia (17%), and Ethiopia (7%), as compared to Kenya (69%) and Zimbabwe (67%). The mutation rate of 184F varied widely from 23–90% across the different countries. For *pfdhps*, the 437G mutant allele prevalence in Ghana (97%) and Kenya (99%) were very high compared to the others (13–26%). By contrast, while the 540E mutation was dominant in Kenya (90%) and Gambia (100%), it was only found in 1% of the samples in Ghana and 15% in Zimbabwe. For *Pfdhps*, the 51I, 59R, and 108N mutant alleles were highly prevalent in Ghana and Kenya compared to Gambia, Ethiopia, and Zimbabwe where a lower percentage of samples showed mutation (Table 3). 

The frequencies of single, triple, quadruple, and quintuple mutations were compared among regions (Table 2 and Table 4; Appendix A). The mutation at *pfdhfr* S108N was defined as a single mutant, *pfdhfr* N51I, C59R, and S108N as triple mutants, *pfdhfr* N51I, C59R, S108N and *dhps* A437G as quadruple mutants, and the combination of *pfdhfr* N51I, C59R, S108N plus *pfdhps* A437G and K540E as quintuple mutants. In the northern savannah (KG and PZ), 97 out of 99 samples (98%) had the *pfdhps* single mutant, 70 (71%) had the *pfdhps* triple mutants, and 45 (45%) had the *pfdhfr*/*dhps* quadruple mutants (Table 4). In the central region (KD), 49 out of 52 samples (94%) had the *pfdhps* single mutant, 43 (83%) had the *pfdhps* triple mutants, and three (6%) had the *pfdhfr*/*dhps* quadruple mutants. In the southern coastal region (AD and DO), 58 out of 59 samples (98%) had the *pfdhps* single mutant, three (5%) had the *pfdhps* triple mutants, and one (2%) had the *pfdhfr*/*dhps* quadruple mutants. Among sites, there was no significant difference for the *pfdhps* single mutant S108N (one-tailed *t*-test; *p*-value > 0.05). However, the prevalence of the *pfdhfr* N51I, C59R, S108N triple mutant was significantly higher in the north and central regions compared to the south (*p* < 0.001; Appendix A). Likewise, a significant difference was observed in the prevalence of the *pfdhfr* N51I, C59R, S108N and *Pfdhps* A437G quadruple mutants between the north and other regions (*p* < 0.001; Figure 3). No quintuple mutants were observed in all samples.

## 4. Discussion

Successful malaria control and elimination rely on accurate and sensitive methods to measure parasite prevalence. This study specifically compared the sensitivity and specificity of different molecular methods in order to assess the true asymptomatic malaria prevalence and determine the most sensitive test. Microscopy remains the gold standard in malaria case management despite a detection limit of 10–50 parasites/μL [31,32]. By contrast, PCR methods can detect ≤5 parasites/μL and are known to be markedly more sensitive than microscopy [33,34]. Consistent with previous studies, our findings indicated that the sensitivity of 18S rRNA nested PCR was the lowest (63.8%) given the low parasite density in asymptomatic infections. Compared to 18S rRNA nested PCR, qPCR methods detected a larger number of positive samples. Almost all samples detected positive by TARE-2 qPCR were also positive using 18S rRNA qPCR, with the exception of four samples that were detected as positive by TARE-2 and negative by 18S rRNA qPCR. Because of the high-copy nature of the TARE-2 gene, it offers a more sensitive gene marker than 18S rRNA in detecting low-density *P. falciparum* infections among asymptomatic samples in this study [10]. This study reinforces the need of novel gene target for parasite screening of low-density infections, and the TARE-2 gene has so far produced very promising results. 

Asymptomatic prevalence in the northern savannah was found to be relatively high compared to that in the south. Land use and socio-economic factors might explain such differences. In the north, agricultural-related activities such as irrigation throughout the year could increase mosquito habitats and allow high malaria transmission [35]. Previous studies indicated that the entomological inoculation rate (EIR) in the north (e.g., Navrongo) was 643 infective bites per person per year (*ib/p/y*) [36], about 30-times higher than in the south (e.g., Dodowa; 21.9 [37]) where urban communities predominate. The fewer mosquito habitats available in the south may be associated with low transmission. A national survey on key socio-economic factors among Ghanaians conducted in 2016 showed 25% of the population in the Northern Region did not seek medical treatment in the six months duration of the survey. Comparatively, in the Central and the Greater Accra Region, 7–9% of the population did not seek for medicine or medical care in the six months of the survey [38]. Such a contrast indicates that people living in rural areas may have poor access to hospitals or are unable to afford medical diagnosis or treatment. Apart from insufficient healthcare infrastructure in the north, human movements from the north to the south due to resource endowments and labor demand could enhance the spread of malaria [39]. In fact, the southern regions of Ghana are the destinations for 88 percent of all internal migrants, while the northern and two upper regions together account for only 5 percent of the total [40]. These socio-economic factors and migration patterns might play a role in the spread of drug resistance and contribute to the overall malaria burden. In this study, we did not observe a significant difference in parasitemia among sites, although the study sites in the north have a much higher prevalence rate. These disparities could be due to the fact that in high transmission areas, children are routinely exposed to the parasite and develop immunity that enables them to remain asymptomatic. In contrast, individuals who live in low transmission areas have less exposure to the parasites, and this might result in a weak immune system [41]. While immunity offers protection against malaria at the individual level, the asymptomatic population provides a reservoir for the parasites and could have a profound effect to the transmission of the disease.

The low prevalence of *pfcrt* mutant allele across the country corroborates with the withdrawal of CQ as first-line treatment for *P. falciparum* since 2004 [20]. Apart from *pfcrt* codon 76, the overall low prevalence of *pfmdr1* Y86 among our study sites is consistent with the pattern seen in *pfcrt*, suggesting the absence of strong selection pressure against CQ. Previous studies conducted between 2012 and 2016 in Ghana comparing the *pfcrt* K76T and *pfmdr1* N86Y prevalence also showed similar trends with the decline of both mutant alleles [24]. It is likely that reduced selection of CQ-resistance strains has allowed the wild-type phenotype to predominate across Ghana. On the other hand, *pfmdr1* F184 mutation has been shown to be strongly associated with amodiaquine resistance [13,35]. The relatively high prevalence of *pfmdr*1 F184 mutation in the central (KD: 72%) and south (AD: 57% and DO: 44%) is concerning with regard to the efficacy of amodiaquine. Nevertheless, the relatively low prevalence of *pfmdr*1 F184 mutation in the north (KG: 26% and PZ: 36%) suggested that the use of amodiaquine for both chemoprevention and first-line treatments has not yet imposed strong selection in the north.

The frequency of mutation in both *pfdhfr* and *pfdhps* were considerably high ranging from 91% to 100% across all three ecological zones. When single, double, triple, and quadruple mutants were examined, we observed striking differences in the triple and quadruple mutants among regions. The prevalence of triple mutant (*pfdhfr* I51/R59/N108) was significantly higher in the north and central regions than in the south. The quadruple mutant (*pfdhfr* I51/R59/N108 + *pfdhps* G437) was significantly higher in the north (36–54%) than the central and south regions (0–6%). Similar trends have been described in Senegal where a significant increase of the quadruple mutation was reported in a region under SMC between 2003 and 2011 [22]. Our findings suggested that the SMC may have a greater impact on a set of linked mutations at the different gene codons that confer a high level of SP resistance than on a single mutation [22]. It is noteworthy that the SMC may not be the only driver of *pfdhfr* and *pfdhps* polymorphisms in our study sites. In 2003, Ghana adopted the WHO recommendation and implemented the Intermittent Preventive Treatment in pregnancy women (IPTp) with SP as the standard of care in malaria-endemic regions [25]. The IPTp may in part explain the overall high prevalence of *pfdhfr* and *pfdhps* mutations across Ghana, as well as the similar trend in mutation prevalence between Ghana and Kenya [42,43]. High prevalence of the *pfdhfr* or *pfdhps* resistance alleles among *P. falciparum* in Ghana was reported as early as 2003 [44]. It is not surprising to see such a pattern prevails in our samples after more than a decade. While the majority of our samples are known to be polyclonal based on microsatellite data (unpublished), we did not detect a large number of mixed nucleotides at the targeted positions within samples. Thus, it is plausible that different clones within the host have undergone the same mutational changes by similar selection. It is yet to be tested whether parasite gene flow might also play a role in the wide distribution of the *pfdhfr* and *pfdhps* mutations. 

In conclusion, this study highlights the observations of high asymptomatic prevalence and quadruple *pfdhfr*/*pfdhps* mutants in the north. These findings have important implications for the efficacy of ongoing SMC and IPT interventions in Ghana. Although this study did not assess the efficacy of chloroquine in treating clinical falciparum malaria, the resurgence of CQ susceptible genotype poses the possibility of reintroducing CQ as a first-line falciparum malaria treatment, as well as part of the SMC regime. We are currently investigating copy number variation for *pfmdr*1, as well other gene markers to further explore how the parasite genomes alter susceptibility to various antimalarial drugs. Our findings emphasize the need for highly sensitive methods to assess the accurate parasite prevalence especially in high malaria-endemic areas for effective disease control and management.

## Figures and Tables

**Figure 1 genes-10-00538-f001:**
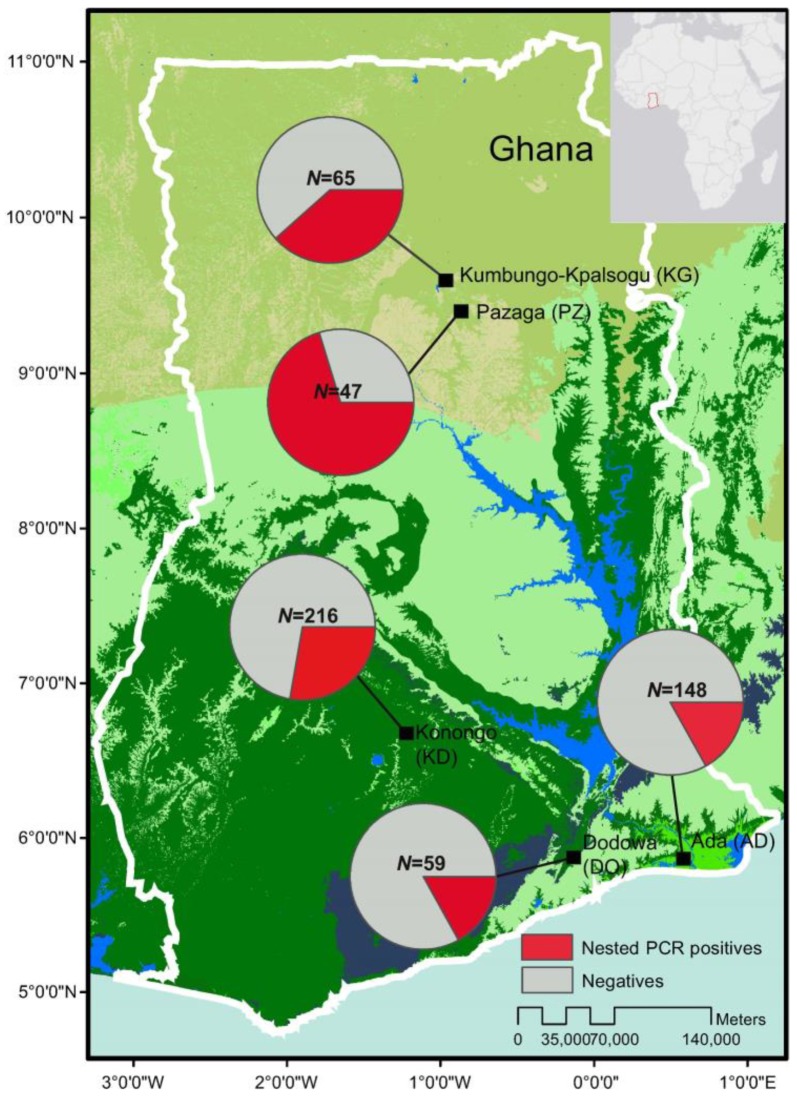
Map showing the study sites in Ghana and the prevalence of asymptomatic *P. falciparum* infections based on 18S rRNA nested PCR (pie charts). *N* indicated the total number of samples collected and screened in each of the study sites (Appendix A).

**Figure 2 genes-10-00538-f002:**
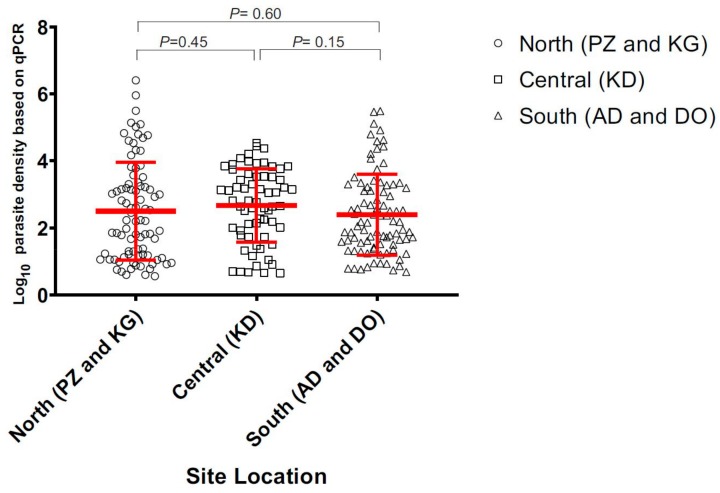
Estimated parasite density of samples collected in the north, central, and south of Ghana.

**Figure 3 genes-10-00538-f003:**
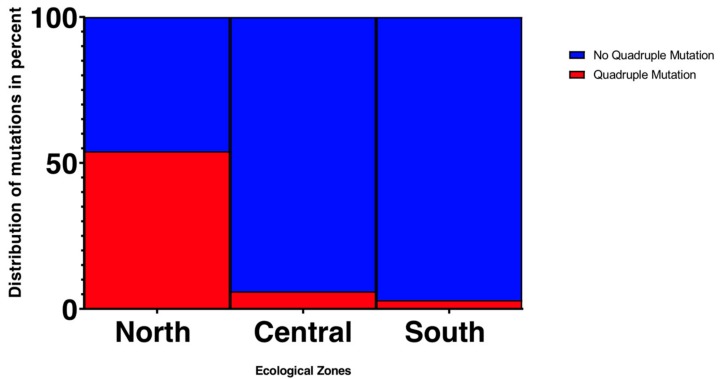
Frequency of the quadruple mutation *pfdhfr* N51I/C59R/S108N, *pfdhps* A437G across the different ecological zones.

**Table 1 genes-10-00538-t001:** Sensitivity and specificity table showing the percentage of positive infections detected by the different methods, sensitivity, and specificity of PCR against microscopy for the 135 samples collected from the north (sites KG and PZ).

Method	Positive	Negative	Sensitivity (95% CI)	Specificity (95% CI)
Microscopy *	35 (18.5%)	100 (81.5%)	-	-
18s Nested PCR	62 (45.9%)	73 (54.1%)	63.8% (53.3–73.5%)	91.3% (71.9–98.9%)
18s qPCR	76 (56.3%)	59 (43.7%)	80.9% (71.4–88.2%)	75.5% (60.4–87.1%)
TARE-2 qPCR	89 (65.9%)	46 (34.1%)	81.2% ((74.4–86.8%)	48.3% (30.1–66.9%)

* Microscopy was used as gold standard.

**Table 2 genes-10-00538-t002:** Prevalence of the *Pfcrt*, *Pfmdr1*, *Pfdhfr*, and *Pfdhps* point mutations in isolates from the three geographical regions (five different sites) in Ghana.

*N*	Gene	Codon		Total (%)					
				**Region**					
				North		Central	South		
				KG	PZ	KD	AD	DO	
	***Pfcrt***								
165		K76T	K	38 (97)	36 (100)	45 (98)	26 (96)	16 (94)	161 (99)
			T	1 (3)	0	1 (2)	1 (4)	1 (6)	4 (1)
	***Pfmdr1***								
216		N86Y	N	55 (100)	61 (94)	45 (100)	29 (88)	17 (94)	207 (96)
			Y	0	4 (6)	0	4 (12)	1 (6)	9 (4)
183		Y184F	Y	40 (74)	32 (64)	13 (28)	6 (43)	10 (56)	101 (55)
			F	14 (26)	18 (36)	34 (72)	8 (57)	8 (44)	82 (45)
	***Pfdhps***								
109		A437G	A	3 (9)	0	0	0	0	3 (3)
			G	30 (91)	33 (100)	45 (100)	24 (100)	13 (100)	106 (97)
155		K540E	K	32 (100)	40 (98)	45 (100)	24 (100)	13 (100)	154 (99)
			E	0	1 (2)	0	0	0	1 (1)
	***Pfdhfr***								
189		N51I	N	6 (12)	4 (8)	6 (12)	4 (11)	0	20 (11)
			I	43 (88)	47 (92)	43 (88)	33 (89)	0	169 (89)
209		C59R	C	13 (28)	12 (23)	4 (8)	0	0	29 (14)
			R	34 (72)	41 (77)	47 (92)	35 (100)	23 (100)	180 (86)
205		S108N	S	2 (4)	0	3 (6)	0	0	5 (2)
			N	45 (94)	52 (100)	49 (94)	35 (100)	23 (100)	204 (98)

*N* denoted the total number of successfully amplified samples. Percentages (%) were presented in parentheses.

**Table 3 genes-10-00538-t003:** Comparison of prevalence of the *Pfcrt*, *Pfmdr1*, *Pfdhfr*, and *Pfdhps* point mutations in *P. falciparum* from West (Ghana from this study; Gambia [27]), East (Ethiopia [28]; Kenya [29], and Southern Africa (Zimbabwe [30]).

Gene	Codon						
			**West Africa**			**East Africa**	**Southern Africa**
			Ghana	Gambia	Ethiopia	Kenya	Zimbabwe
***Pfcrt***							
	K76T	K	161 (99)	138 (57)	116 (39)	69 (27)	22 (20)
		T	4 (1)	105 (43)	189 (61)	185 (73)	90 (80)
***Pfmdr1***							
	N86Y	N	207 (96)	202 (83)	142 (93)	78 (31)	37 (33)
		Y	9 (4)	43 (17)	10 (7)	174 (69)	75 (67)
	Y184F	Y	101 (55)	88 (36)	18 (10)	195 (77)	-
		F	82 (45)	157 (64)	181 (90)	58 (23)	-
***Pfdhps***							
	A437G	A	3 (3)	180 (74)	152 (85)	3 (1)	98 (87)
		G	106 (97)	63 (26)	27 (15)	250 (99)	14 (13)
	K540E	K	154 (99)	23 (10)	-	0 (0)	96 (85)
		E	1 (1)	222 (90)	-	253 (100)	16 (15)
***Pfdhfr***							
	N51I	N	20 (11)	240 (98)	45 (23)	8 (3)	100 (89)
		I	169 (89)	5 (2)	154 (77)	245 (97)	12 (11)
	C59R	C	29 (14)	241 (98)	100 (50)	26 (10)	102 (91)
		R	180 (86)	6 (2)	99 (50)	227 (90)	10 (9)
	S108N	S	5 (2)	243 (99)	44 (22)	0 (0)	72 (64)
		N	204 (98)	2(1)	155 (78)	253 (100)	40 (36)

*N* denoted the total number of successfully amplified samples. Percentages (%) were presented in parentheses. ‘-’indicated data not available.

**Table 4 genes-10-00538-t004:** Haplotype prevalence of mutations in the *pfdhfr* codons N51I, C59R, S108N, and *pfdhps* A437G and K540E present among the *P. falciparum* isolates from three regions in Ghana. *N* denoted the total number of successfully amplified samples. Percentages (%) were presented in parentheses.

Region	Study Site		*N* (%)			
			*Pfdhps*	*Pfdhfr*	*Pfdhfr/Pfdhps*	*Pfdhfr/Pfdhps*
		Total	N108	I51R59N108	I51R59N108/G437	I51R59N109/G437E540
North						
	KG	47	45 (96)	33 (70)	17 (36)	0
	PZ	52	52 (100)	37 (71)	28 (54)	0
Central						
	KD	52	49 (94)	43 (83)	3 (6)	0
South						
	DO	23	23 (100)	0	0	0
	AD	36	35 (97)	3 (8)	1 (3)	0

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
