# Peer review of "Contrasting Asymptomatic and Drug Resistance Gene Prevalence of Plasmodium falciparum in Ghana: Implications on Seasonal Malaria Chemoprevention"

_genes, 2019, doi:10.3390/genes10070538_

Round 1

Reviewer 1 Report

Summary

The manuscript presents a study that evaluated the sensitivity and specificity of different molecular methods in estimating the prevalence of asymptomatic malaria in Northern Ghana, where there is intense and seasonal malaria transmission.  They also examined the frequency of mutations in pfcrt, pfmdr1, pfdhfr, 17 and pfdhps that they could relate to the ongoing seasonal chemoprevention strategy in Ghana that combines Sulfadoxine-Pyrimethamine and Amodiaquine.  They observed higher prevalence of asymptomatic malaria in the north than in the South. They also reported higher prevalence of the quadruple mutation associated with sulfadoxine-Pyrimethamine resistance in the north. 

Overall.

This is a well written manuscripts, with clear description of methods, clear presentation of results. However, the discussion could have been strengthened by provide references to support statements. Moreover, conclusions were not clearly stated in the manuscript.

Specific comments.

1.    It is not clear the rationale or motivation of using several molecular tests to determine asymptomatic malaria prevalence? Is the goal to find the most sensitive test? Please clarify? Also, it would have made sense to test by microscopy first – then test microscopy positives and negatives by the molecular methods. 

2.    Assessing sensitivity and specificity requires that a new assay or method is tested against a well-established method. Although you state that TARE-2 is very highly sensitive method (and for this reason you used it as gold standard), I still believe that the gold standard should be microscopy as it is a widely used method for malaria testing. I appreciate that all methods have their drawbacks but under reasonable conditions, it would be more sensible to  assess your different tests against  microscopy as the gold standard.

3.    Resistance mutation testing has been explained very well. Good!

4.    The first sentence in the results section” Of the 535 total samples, 105 (19.6%) were found positive by microscopy.” What about the numbers positive by the other methods - 18S rRNA nested PCR, 18S rRNA qPCR, and TARE-2 qPCR. Please state them as well.

5.    In paragraph 2, in explaining the differences in asymptomatic prevalence in the north and South, the authors point to the fact that mosquito densities may be high due to the irrigation activities in the north. However, there is no evidence or further support of this statement.

6.    Paragraph 3 could be merged with paragraph 2.

Reviewer 2 Report

p.p1 {margin: 0.0px 0.0px 0.0px 0.0px; line-height: 15.0px; font: 13.3px Arial; color: #000000; -webkit-text-stroke: #000000; background-color: #f4f4f4} p.p2 {margin: 0.0px 0.0px 0.0px 0.0px; line-height: 15.0px; font: 13.3px Arial; color: #000000; -webkit-text-stroke: #000000; background-color: #f4f4f4; min-height: 15.0px} span.s1 {font-kerning: none}

The manuscript “Contrasting asymptomatic and drug resistance gene prevalence of Plasmodium falciparum in Ghana: implications on Seasonal Malaria Chemoprevention” describes data collected from ongoing efforts for malaria prevention in Ghana. The analysis of the 535 samples collected from 5 locations was thorough, including several different malaria testing methods and analysis of resistance genes. 

- The discussion looks at resistance alleles from previous studies in other countries. If the data is available, it would be interesting to add these numbers from other countries to Tables 2 and 3, so the regions can be directly compared to the results from other countries. Even if there are data points missing from the other countries, this would help to better frame the results in context with the literature. 

- Figure 1: The very dark colors on the pie charts make them difficult to read. They should be brightened up. 
